# A Behavioral Strategy to Nudge Young Adults to Adopt In-Person Counseling: Gamification

**DOI:** 10.3390/bs12020040

**Published:** 2022-02-07

**Authors:** Shengen Piao, Jaewoo Joo

**Affiliations:** 1Graduate School of Techno Design, Kookmin University, 77 Jeongneung-ro, Seongbuk-gu, Seoul 02707, Korea; sung-en@hotmail.com; 2Department of Marketing, College of Business Administration, Kookmin University, 77 Jeongneung-ro, Seongbuk-gu, Seoul 02707, Korea

**Keywords:** gamification, adoption, usability, vividness, counseling, nudge

## Abstract

Mental illness has always been an important issue for young adults. Moreover, initiatives resulting from the outbreak of COVID-19 have had an even greater impact on the mental health of young adults. This study sought to examine the effect of gamification on whether young adults adopt in-person counseling. One hundred twenty young adults (42 males and 78 females) with an average age of 29 years participated in our experiment. In the experiment, a 2 (Gamification: no vs. yes) × 2 (Vividness: low vs. high) between-subjects design was employed. In the “yes” gamification condition, participants decided whether or not to read introductory material about in-person counseling, and also whether or not to adopt in-person counseling in the future. The results of the study show that: (1) gamification increased adoption, (2) participants’ perception of subjective usability of in-person counseling mediated the effect of gamification to adoption, and (3) vividness of presentation moderated subjective usability. Our study demonstrated that gamification nudges young adults to adopt in-person counseling while subjective usability mediates the relationship, and vividness moderates the relationship between gamification and subjective usability. Our findings provide counselors fresh insights into motivating people to access counseling services.

## 1. Introduction

According to the World Health Organization, up to 20% of children and adolescents have a mental illness [1]. Half of these illnesses begin before the age of 14, and most cases go undetected and untreated [2]. According to the US Centers for Disease Control and Prevention, adverse childhood experiences are “strongly related to the development of risk factors for disease, and well-being throughout the life course” [3]. Suicide is the fourth leading cause of death among those between the ages of 15 and 29 [4]. People with depression, a form of mental illness, are 25 times more likely to commit suicide than those who are not experiencing depression [5]. Analysis of the suicide rate in high-income countries shows that up to 90% of suicidal cases are related to mental illness. Furthermore, almost 76% to 85% of people with depression remain untreated in low- and middle-income countries [2,4]. According to the World Bank and the World Population Review, South Korea is a high-income country that has the fourth highest suicide rate in the world, while China is a upper-middle-income country with the fifth leading cause of death by suicide, accounting for more than a quarter of the global suicide population [6,7].

To address mental illness, we have to pay closer attention to the behavior of people seeking help from professionals, and more precisely, the accessibility of in-person counseling. Note that mental illness may look like a cold that people rarely think about seriously [8]. Therefore, when people suffer from mental illnesses, they may not seek psychologists’ professional help but merely talk to someone they can trust, such as friends or family [9,10]. Indeed, seeking professional help is not only complicated but also unnatural for some people. They may be concerned about being stigmatized, fearing that they will be perceived as spiritual freaks [11]. As the stigma of mental illness increases, people become increasingly resistant to seeking professional help [12]. Eastern countries have a particularly higher level of stigma and more moral attributions than Western countries, especially regarding depression [13].

Although many attempts have been made to help people to adopt in-person counseling, previous studies have focused only on improving the original form of counseling or providing telemedicine counseling. For example, one study used behavioral activation to increase positive emotions based on in-person counseling, termed positive psychotherapy [14]. Mazzucchelli et al. concluded that positive psychologists and mental health counselors could readily adopt behavioral activation to promote well-being [15]. However, due to COVID-19, office activities have taken a hit, which led to speculation that counseling will most likely become a hybrid, offered both online and offline [16]. In the area of telemedicine counseling, most studies have discussed the management of chronic diseases, particularly diabetes. Benhamou suggested that telemedicine can improve the treatment of Type 2 diabetes, but the application to Type 1 diabetes remains ambiguous [17]. Ronda and Dijkhorst-Oei noted the effective use of telemedicine in treating diabetes, but it was disappointing to encounter patients who were apathetic about the management of their disease [18]. Kruse and Argueta showed that significant improvements in self-management of patients with chronic diseases and the quality of care provided by providers can be made but negative perceptions about security and usability were still a concern [19].

We propose a new strategy, gamification, to help young adults to adopt in-person counseling. Young adults are in the midst of an important event called “relationships”, so a good mental health is crucial for individuals and society as a whole. The next stage of young adults corresponds to “Work and Parenthood”, and thus, poor mental health may cause bad care and affect their next generation [20,21]. Gamification is rapidly becoming a trend within the health professional education industry. It has been recently implemented in medical education and is increasingly used as an adjunct to traditional teaching strategies [22]. In academia, two previous publications discuss the impact of gamification on mental health. One is Litvin et al. who developed a game called eQuoo and tested its impact on psychological resilience, well-being, and anxiety attrition rates [23]. The other is Poppelaars et al., who used a mobile game, Monument Valley, to test whether mental health messages can engage people with mental health symptoms, concluding that the game can increase people’s well-being [24]. Among practitioners, gamification of mental health services is also starting to emerge, for example, Tripp’s VR-wellness platform offers an “eye-opening meditation” app in VR [25]. It provides a customized meditation service based on the meditator’s mood and time zone. A meditation platform called Headspace offers fun and simple meditations, providing an online service where people can experience the benefit of meditation anytime, anywhere [26].

Although gamification attracts significant attention among researchers and practitioners, three issues still remain unclear, in particular concerning whether the reported gamification effects can be attributed solely to their intervention. Firstly, most studies were descriptive [27,28,29]. They did not conduct experiments but reported findings based on the observations of what has already been done. Secondly, as van Gaalen suggests, most research fails to establish any underlying process [30]. Therefore, it remains unclear why particular game attributes or combinations benefit people. Lastly, although some researchers addressed previous issues by conducting experiments, these exceptional papers tested the gamification effect only on adolescents, concluding that the same effect has not been tested rigorously among adults [31].

Therefore, we aim to address these issues by testing to see if a game of in-person counseling encourages young adults to adopt in-person counseling. To test our Hypothesis, we conducted an experiment by employing a between-subject design. More specifically, we developed a virtual world-based game of the Psychological Diagnostic Institute and collected participants’ responses toward the virtual game as well as in-person counseling. In this study, we conceptualized young adults’ adoption of in-person counseling as a dependent variable, gamification as an independent variable, objective and subjective usability as a mediating variable, and the vividness of presentation as a moderating variable. Through the experimental study, we found a positive effect of gamification on adoption of in-person counseling, a mediating effect of subjective usability on adoption of in-person counseling, and a different effect of vividness on objective and subjective usability of in-person counseling.

## 2. Literature Review

### 2.1. Gamification

Gamification motivates learning by either directly using games or indirectly transferring game elements to non-game contexts [32]. The term “gamification” was first introduced in 2002 by Nick Pellin, a British-born computer programmer and game developer [33]. In general, gamification begins when purposeful game elements are introduced to promote learner’s engagement, motivation, and behavior change [34].

Many researchers have proposed the various game elements. Some suggest that frequently observed game elements in the non-game environment are points, badges, and leaderboards [35]. Others identified thirty-nine different game elements through a taxonomy of game attributes [36]. Nineteen game attributes were defined to comprehensively describe the whole existing game elements [37]. Researchers recently reached a consensus that there are nine independent game elements [38] which are comprised of action language, assessment, conflict/challenge, control, environment, game fiction, interpersonal interaction, immersion, and rules/goals.

Note that gamification researchers have paid unequal attention to the nine game elements. In particular, van Gaalen pointed out that prior research on gamification highlighted assessment and conflict/challenge extensively [30]. For instance, Bhaskar selected assessment, environment, immersion, and rules/goals to examine the effect of gamification on blood type education [39]. While investigating the effect of gamification on evaluating surgical decisions, Lin et al. selected conflict/challenge and rules/goals [40]. El-Beheiry et al. selected action language, assessment, conflict/challenge, and immersion to validate the effect of simulated course gamification on the manipulation and proficiency of the apparatus [41]. Indeed, van Gaalen suggested future research should explore theories that can explain the impact of gamification interventions and explore them in a longitudinal approach with an explicit control group [30].

Thus, we selected environment, one of the nine game elements which has not been well-studied. Bedwell et al. used a card sorting approach in which the location element was redefined as the environment element [38]. According to Owen’s the definition of location is the physical or virtual world in which the game is played [42]. Therefore, we decided to make a game to introduce in-person counseling, transferring the counseling room to a three-dimensional virtual world as shown in Table 1.

### 2.2. Adoption

New product adoption is defined as the “acceptance and continued usage of a new product by a consumer” [43]. Joo shows that when consumers engage in adoption behavior, they are more likely to prefer to adopt products that reflect their known consumer utility [44]. However, when products involve consumers’ blind spots, adding information can increase consumers’ product attitudes [45]. At the same time, the user’s own ability to conceptualize information also determines the adoption of new products [46]. Those results indicate that the adoption of new products should not only consider what information is presented to consumers, but also whether the information presented to consumers can be understood.

New products can be defined as “original products, product improvements, product modifications, and new brands” [47]. Here, product can be defined as something that “includes more than just tangible objects…. Broadly defined, products also include services, events, persons, places, organizations, ideas, or a mixture of these” [47]. Since no service such as the gamification of in-person counseling exists, we can equate adopting the gamification of in-person counseling as a new product adoption.

Rogers identified five stages of innovation: Knowledge, Persuasion, Decision, Implementation, and Confirmation [48]. In the decision phase, people choose whether they want to adopt or reject the innovation. This means that adoption is a stage of perception and intention for the pre-consultation of a new product. Our study focused on the first three parts of the model shown in Figure 1.

**Hypothesis 1 (H1).** *Gamification of in-person counseling will increase the adoption of in-person counseling*.

### 2.3. Usability

Usability is defined as “the extent to which a system, product or service can be used by specified users to achieve specified goals with effectiveness, efficiency and satisfaction in a specified context of use” [49]. Usability is a core term in human-computer interaction (HCI). It is formed by the interaction between the tool, the problem, and the person [50].

Evaluating usability is a method that confirms whether the interactive system is adapted to the user [51]. According to the British Design Council’s Two Diamond Model, usability evaluation is an intermediate aspect of the overall stage, which means that usability evaluation is not an outcome but should be a process. However, we found few studies that examined usability as a mediating variable. Researchers have mainly studied it as an independent variable [52,53,54]. Therefore, we proposed that usability mediates the relationship between gamification and adoption.

Note that usability cannot be measured by itself [49]. We broke down usability into two separate concepts, objective and subjective usability. Prior research already provides a comprehensive view of Health Information Technology Usability Evaluation [55]. The classification of usability into objective and subjective usability has been used extensively in previous studies [56,57]. Such a distinction is thought to simplify the nature of the measurement [56,58]. The Yen’s Framework of Usability is shown in Figure 2.

**Hypothesis 2a (H2a).** *Gamification of in-person counseling will increase the objective usability of in-person counseling*.

**Hypothesis 2b (H2b).** *Gamification of in-person counseling will increase the subjective usability of in-person counseling*.

**Hypothesis 3a (H3a).** *Objective usability of in-person counseling will increase the adoption of in-person counseling*.

**Hypothesis 3b (H3b).** *Subjective usability of in-person counseling will increase the adoption of in-person counseling*.

### 2.4. Vividness

A long-held belief in marketing communications is that vivid messages are more persuasive than abstract ones [59,60]. “Information may be described as vivid, that is, as likely to attract and hold our attention and to excite the imagination, to the extent that it is: (a) emotionally interesting, (b) concrete and imagery-provoking, and (c) proximate in a sensory, temporal or spatial way” [59]. Researchers argue that vivid information has a positive effect on people’s attitudes. Visual materials seem to be particularly memorable because of their vividness [61].

Vividness has been found to help to improve the persuasiveness of information provided by others. Smith and Schaffer demonstrated that a message was more persuasive if the vivid image is consistent with the message’s central theme but not significant if the vivid image contradicts the central theme [62]. Guadagno, Rhoads, and Sagarin suggested that vividness strongly influences message success and may positively or negatively impact persuasive communication [63]. However, some studies suggested that any attempt to add vividness fails to increase persuasion [64,65]. Taylor and Thompson suggested that the persuasive effect of vividness may only be apparent under different attention conditions [66]. Under conditions where people’s attention is consistent with the presented material, vivid and unvivid information affects judgment. However, under conditions in which people’s attention is inconsistent with the presented material, vividly presented information may compete more successfully for attention. Thus, under conditions of greater attention, vividly presented information will be more persuasive than non-vividly presented information.

Therefore, we propose that vividness moderates the relationship between gamification and objective and subjective usability. Unlike traditional studies that examined the differences between the same static information (graphic information and textual information) or the same dynamic information (video information and verbal information), recent studies have examined the impact of vividness for static graphic information versus dynamic video information [62,63,64,65,67,68]. Following the recent approach, we investigated the impact of vividness between static graphic information and dynamic video information.

**Hypothesis 4a (H4a).** *The vividness of presentation of in-person counseling will increase the gamification of in-person counseling to objective usability of in-person counseling effect*.

**Hypothesis 4b (H4b).** *The non-vividness of presentation of in-person counseling will not increase the gamification of in-person counseling to the objective usability of the in-person counseling effect*.

**Hypothesis 5a (H5a).** 
*The vividness of presentation of in-person counseling will increase the gamification of in-person counseling to subjective usability of in-person counseling effect.*


**Hypothesis 5b (H5b).** *The non-vividness of presentation of in-person counseling will not increase gamification of in-person counseling to subjective usability of in-person counseling effect*.

## 3. Research Goal

We propose the following psychological processes that increase young adults’ adoption of in-person counseling: (1) The gamification format influences young adults’ understanding of in-person counseling and nudges them to adopt in-person counseling. (2) The effects of gamification on objective and subjective usability via different forms of vividness were observed. (3) We examine the factors that influence the adoption of in-person counseling by young adults across objective and subjective usability.

Previous research confirmed that gamification created a higher quality experience and dynamic information had higher vividness [67,69]. We propose that young adults prefer dynamic information (vividness: high) to static information (vividness: low). The vividness of information, in turn, affects objective and subjective usability, which ultimately impacts the adoption of counseling.

Overall, we present seven Hypotheses regarding the gamification of in-person counseling services to judge young adults’ intention to adopt in-person counseling. These hypotheses are presented above and are also depicted in the conceptual model shown in Figure 3.

## 4. Study

### 4.1. Method

#### 4.1.1. Study Ethics

We based this study on a 2 × 2 between-subjects design. The study was conducted after obtaining ethical approval from the Institutional Review Board of Kookmin University (KMU). The study was conducted from August 2021 to December 2021.

#### 4.1.2. Participants

In psychology, the definition of young adults comes from Erik Erikson’s Stages of Psychosocial Development theory [20]. Although the age range of young adults is not specified in Erik, Kendra Cherry believes that young adults should be between 19 and 40 years old, so we screened our user study group for people in this range [21].

We recruited young adults from two countries: a Korean invitation was posted on Naver Cafe, an online community in Korea, and a Chinese invitation on WeChat Moment, a social media platform in China. The experiment was carried out online on zoom, and we negotiated the schedule with the participants.

In total, 120 young adults from Korean- and Chinese-speaking regions participated in this study. Out of 120 participants, 51.7% came from Korea, 48.3% came from China, 35% were identified as male and 65% as female. The age of the participants ranged from 20 to 40 years old (M_age_ = 29.333, SD_age_ = 4.198), which matches the age criteria for young adults.

#### 4.1.3. Experimental Design

The study was implemented as an experiment following a 2 × 2 between-subjects design, as shown in Table 2. Participants were randomly assigned to one of four groups. Each participant was asked to imagine that they had begun the act of seeking help because of psychological stress and then rate the service they were using. Two factors were manipulated in the experiment: Factor 1: Environment of presenting information (manipulated at two levels: no gamification vs. yes gamification). Factor 2: Form of presenting information (manipulated at two levels: low vividness vs. high vividness).

In the no gamification condition, participants were asked to view full-screen in-person counseling-related presentations provided by the experimenter directly through the zoom remote control function. In the yes gamification condition, participants were asked to view full-screen counseling-related presentations provided by the experimenter in a virtual world through the Zoom remote control function. Participants were free to control their behavior during the experiment, including the ability to stop viewing the material or terminate the experiment. While we manipulated gamification, we took extra care so that participants in each condition were equally informed about in-person counseling. Therefore, the total amount of information participants received in two different conditions was identical.

In the low vividness condition, participants were provided with a static presentation which was an introduction to in-person counseling, while in the high vividness condition, participants were provided with a dynamic video. Note that the two vividness conditions are divided into static information (single frame) and dynamic information (video). Adding audio to the dynamic information may have diminished the persuasion of vividness and ultimately affect the fairness of both materials [70,71]. Therefore, we eliminated any sound in the video material. Both sets of experiments were conducted in a quiet setting.

#### 4.1.4. Procedure

We conducted our experiment online. Due to the COVID-19 lockdown, participants are somewhat resistant to offline gatherings, so we chose to use zoom for our online experiment.

In the online experiment, participants were not notified that the study was designed for gamification related to the adoption of in-person counseling, and participants were unaware of the specific research objectives of the study. After confirmation of informed consent (e.g., regarding the procedure and duration of the study, data protection and use, the voluntary nature of participation, etc.), the experimenters divided the participants into four groups based on average age. All participants were connected to the Zoom meeting room, and each experiment was conducted with one experimenter and one participant. After entering the Zoom room, the experimenter gave the participants access to the remote-control screen. The participants were asked to read and evaluate the in-person counseling-related presentations provided by the experimenter, imagining that they were learning about the introduction of in-person counseling for the first time. Each of the four groups read and evaluated different materials, and there was no time limit on the experiment. Finally, appreciation was extended to the participants, and they were notified about the purpose of the study. Participation in the study took approximately 10–15 min. See Appendix A for questions and scales.

#### 4.1.5. Measures

Adoption. We measured adoption by asking one question (how much do you want to adopt in-person counseling). This question was answered using a Likert scale with seven response options, from Strongly disagree (1) to Strongly agree (7) [72]. Miller believes that people’s minds have a span of absolute judgments that can be distinguished into seven categories and a span of attention that can include six objects at a glance [73]. Moreover, the 7-point scale has a stronger correlation with the results of a *t*-test [74].

Effectiveness. Effectiveness was measured by recall of the objective answer with the correct answer [75]. The questions were extracted from materials introducing in-person counseling. There were five objective questions asked, and the answers were mentioned in the materials introducing in-person counseling. When the correct answer was selected, we determined the recall rate to be 100%. Conversely, it is 0%.

Efficiency. Efficiency was measured by the precision of the subjective answer with the correct answer [76]. The questions are extracted from materials introducing in-person counseling. There is one objective question asked, and the answers are mentioned in the materials introducing in-person counseling. We determined precision to be 100% when the keywords of the correct answer were represented in the subjective answer. Conversely, it is 0%.

Satisfaction. The SUS score measured satisfaction because it allows comparison across environments and systems and can objectively measure satisfaction [77]. Ten items calculated the SUS score using a Likert scale with five options for respondents [78]. The ten survey items are: (1) “I think that I would like to use this system frequently.“ (2) “I found the system unnecessarily complex.“ (3) “I thought the system was easy to use.” (4) “I think that I would need the support of a technical person to be able to use this system.” (5) “I found the various functions in this system were well integrated.” (6) “I thought there was too much inconsistency in this system.” (7) “I would imagine that most people would learn to use this system very quickly.” (8) “I found the system very cumbersome to use.” (9) “I felt very confident using the system.” (10) “I needed to learn a lot of things before I could get going with this system.”

#### 4.1.6. Data Analysis

First, we mainly used Hayes’ PROCESS macro in this experiment. Hayes’ statistical method is an emerging method to validate Moderator and Mediator, and it has gradually been used in recent marketing studies and design studies [46,79]. Second, in order to verify the credibility of Hayes’ method, we performed another validation using traditional methods. Although the results obtained may be the same, the principles of the two methods are different. We felt it was necessary to test the hypothesis using both methods to make the experiment more rigorous. We describe each analysis in detail below.

We used an independent samples *t*-test to find the relationship between gamification and the adoption of in-person counseling to test whether gamification would increase the adoption of in-person counseling among young adults.

We used Hayes Model 4 to identify and explicate the relationship between gamification and adoption of in-person counseling that may be affected via the interaction of objective usability and subjective usability to test the mediating effects of objective usability and subjective usability on gamification to adoption of in-person counseling.

We used Hayes Model 7 to explore the effect of gamification on the adoption of in-person counseling via respondents’ perceptions of objective usability and subjective usability at different values of presentation vividness of in-person counseling to test the moderating effect of vividness on gamification to objective and subjective usability. Secondly, we used a two-way AVONA to re-evaluate these two moderating effects.

### 4.2. Results

Data collected through surveys on the adoption of in-person counseling were analyzed using the SPSS 26.0 statistical program and G*Power 3.1 program. Our hypotheses mainly consist of two parts: Part A examined the mediating effect of perceptions of gamification’s objective and subjective usability on the adoption of in-person counseling. Part B examined the moderating mediation effect of the vividness of presentation and gamification on the objective and subjective usability of in-person counseling. To test the effect of mediation and moderated mediation, we used conditional process modeling (PROCESS macro) for SPSS [80]. This tool allowed us to perform mediation (Hayes Model 4) and moderated mediation (Hayes Model 7) tests to assess the indirect effects of gamification (no gamification and yes gamification) on the adoption of in-person counseling via the mediating mechanisms of usability at different levels of vividness (low vividness and high vividness). Moreover, to measure the reliability and validity of the experimental classification of usability into objective and subjective, we conducted an Exploratory Factor Analysis (EFA). In addition, we treated sex, age, and nationality as independent covariates.

#### 4.2.1. Demographic Information about Participants

We collected responses from 120 participants. A summary of the descriptive data is shown in Table 3. We collected data from 42 males (35%) and 78 females (65%). The average age of the participants was 29.333 (SD_age_ = 4.198).

Regarding whether the different nationalities of the participants bring additional variables to the research, we found that the nationality of the same cultural area has no effect on gamification to the adoption of in-person counseling (M_China_ = 5.24, M_Korea_ = 5.48, F = 0.002, df = 1, *p* = 0.962).

#### 4.2.2. Reliability and Construct Validity

The internal reliability and validity of the objective and subjective usability variables were measured by exploratory factor analysis, as shown in Table 4, Table 5 and Table 6. Although the eigenvalue of component 2 did not exceed 1, we decided to use two components to explain usability because component 1 only explained 57.14% of the sample, and the correlation between effectiveness and efficiency was significant within 0.1%. Component 1 can be interpreted as objective usability, and component 2 can be interpreted as subjective usability.

#### 4.2.3. Testing Hypotheses

Hypothesis H1 was supported by independent sample *t*-tests that showed that gamification positively affects adoption of in-person counseling within a significant value of 10% (t = −1.820, df = 118, *p* = 0.071). The average score for the adoption of in-person counseling among the non-gamification participants in this study was 5.170 ± 1.224 (mean ± SD). The average score for the adoption of in-person counseling among the gamification participants in this study was 5.570 ± 1.184 (mean ± SD). The statistical power of sample size of gamification on adoption is shown in Table 7.

Hypotheses H2a and H2b were supported with the help of Hayes’s mediation model (Model 4), demonstrating that gamification has a positive effect on the objective and subjective usability of in-person counseling by a significant value of 10%. The result concerning the direct effects is shown in Table 8. The direct effect of gamification on the objective usability of in-person counseling (path a = 0.546) was entirely above zero (90% CI with LL = 0. 246 and UL = 0. 861), the effect of gamification on objective usability was significant at *p* < 0.10 (no zero included in the 90% CI); The direct effect of gamification on the perceived subjective usability of in-person counseling (path a = 0.434) was entirely above zero (90% CI with LL = 0. 130 and UL = 0.738), the effect of gamification on objective usability was significant at *p* < 0.10 (no zero included in the 90% CI).

Hypothesis H3b was supported by Hayes’s mediation model (Model 4), which showed that the perception of subjective usability positively affects the adoption of in-person counseling at a significant value of 10%. The result concerning the direct effects is shown in Table 8. The direct effect of subjective usability on adoption of in-person counseling (path b = 0.426) was entirely above zero (90% CI with LL = 0.247 and UL = 0.604), and the effect of gamification on objective usability was significant at *p* < 0.10 (no zero included in the 90% CI).

Table 9 represents the indirect effect of gamification on the adoption of in-person counseling via the perceived usability of in-person counseling. We used bias-corrected bootstrapping that used 5000 bootstrap samples to examine the indirect effect of gamification on the adoption of in-person counseling via the usability of in-person counseling. The indirect effect of gamification on the adoption of in-person counseling via the subjective usability of in-person counseling was entirely above zero (bootstrapped indirect effect = 0.185; 90% CI with BootLL = 0.042 and BootUL = 0.372). The statistical power of sample size is shown in Table 10.

Hypotheses H5a and H5b were supported with the help of Hayes’s moderated mediation model (Model 7). The results show that the interaction effect of gamification and presentation vividness positively affects the subjective usability of in-person counseling at a significant value of 10%. The result of the interaction effects is shown in Table 11. Table 12 depicts the conditional effect of vividness on the subjective usability of in-person counseling (α = 1.079, *p* = 0.003). Here, alpha is the coefficient showing the strength of the relationship between vividness and the subjective perception of the usability of in-person counseling. The X values correspond to the absence (no gamification = 0) and presence (yes gamification = 1) effects of the independents; W values correspond to the absence (low vividness = 0) and presence (high vividness = 1) effects of the moderators. The conditional effect included zero (90% CI with LL = −0.477 and UL = 0.327 at W = 0) when in the low vividness condition, the effect of low vividness on subjective usability was not significant at *p* < 0.10 (zero is included in the 90% CI); but was significant in the high vividness condition (90% CI with LL = 0.578 and UL = 1.428 at W = 1). The effect of high vividness on subjective usability was significant at *p* < 0.10 (no zero was included in the 90% CI).

Table 13 denotes that the indirect effects of gamification on the adoption of in-person counseling via subjective usability are positive and increase with the vividness associated with a presentation of in-person counseling. We used bias-corrected bootstrapping that used 5000 bootstrap samples to examine the indirect effect of gamification on the adoption of in-person counseling via a moderated mediation effect. The indirect effect of gamification on the adoption of in-person counseling via moderated mediation was entirely above zero (bootstrapped indirect effect = 0.427; 90% CI with BootLL = 0.178 and BootUL = 0.732 at W = 1) when in the high vividness condition, but not significant at 0 for interaction activity (bootstrapped indirect effect =−0.427; 90% CI with BootLL = −0.199 and BootUL = 0.155 at W = 0).

In addition, we performed a Two-Way ANOVA to re-evaluate H4(a,b) and H5(a,b). We found a significant interaction effect of gamification and vividness on the objective and subjective usability of in-person counseling, as shown in Table 14. According to the Two-Way ANOVA method, vividness had a significant positive effect on subjective usability (90% CI with LB = −0.721 and UB = −0.139 at X = 0; 90% CI with LB = 0.162 and UB = 0.744 at X = 1) in the high vividness condition; vividness had no significant effect on subjective usability (90% CI with LB = −0.252 and UB = 0.330 at X = 0; 90% CI with LB = −0.353 and UB = 0.229 at X = 1) in the low vividness condition. We concluded that subjective usability completely mediates the association between gamification and the adoption of in-person counseling (H5a and H5b). The result of the mediation analysis is shown in Table 15. The statistical power of sample size of Gamification × Vividness on objective and subjective usability is shown in Table 16.

### 4.3. Discussion

Our study showed that young adults’ adoption of in-person counseling increased under gamification conditions, and demonstrated that young adults’ perception of subjective usability of in-person counseling ultimately influenced their adoption of in-person counseling.

The detailed results are as follows: gamification increases the adoption of in-person counseling (H1); gamification increases the perception of the objective usability of in-person counseling (H2a); gamification increases the perception of subjective usability of in-person counseling (H2b); subjective usability increases the adoption of in-person counseling (H3b); the vividness of presentation increases the subjective usability of in-person counseling (H5a); non-vividness of presentation does not increase the subjective usability of in-person counseling (H5b).

Although the effect of vividness on objective usability was not confirmed (H4a and H4b), we found a negative effect of vividness on objective usability. However, it also proves and complements Guadagno, Rhoads, and Sagarin’s argument that vividness strongly influences message success and may positively or negatively impact persuasive communication [63]. In our case, gamification positively affected the adoption of in-person counseling and objective and subjective usability. Furthermore, vividness moderated the relationship between gamification and both objective and subjective usability.

## 5. General Discussion

### 5.1. Summary

Our experiments supported the hypotheses that (1) When people are provided with gamified in-person counseling, they are more likely to engage in counseling in real life (H1); (2) When people use gamified in-person counseling, the associated objective and subjective usability ratings are improved (H2a and H2b); (3) Subjective usability or degree of satisfaction is also significant for participants’ adoption of in-person counseling (H3b); (4) Participants gave the highest subjective usability ratings in the mediating condition of vividness, and vividness did differ to an extent in the gamified and non-gamified conditions (H5a and H5b).

However, by mediating the effect of reconciliation, we also found findings beyond the hypothesis: the vividness of the presentation (compared to static image material and dynamic video material) harmed objective usability. Since we conducted only one experiment, we cannot define whether this effect comes from vividness.

### 5.2. Academic Contributions

Our findings of the effect of gamification on the adoption of in-personal counseling align with the findings obtained from the prior studies [81,82,83] In the past, for instance, Butt showed that gamification increases the usability of the VR learning system. Van Nuland showed that gamification increases knowledge of educational competition. Finally, Verkuyl showed that gamification increases satisfaction and knowledge about clinical care. A closer look at the data reveals that gamification has a more substantial impact on objective usability than subjective usability (see Table 8). This result is not surprising, as gamification aims to facilitate curricular goals [22].

Our findings on the mediating effects of perceptions of the objective and subjective usability of gamified in-person counseling and thus its adoption is different from prior studies [77]. In the past, for instance, Georgsson evaluated the usability of the mHealth system and found a need for improvement. Still, they lacked a comparison test group, and it was difficult to judge whether the effectiveness of a product can be evaluated in general terms by the usability alone. Our reliability and construct validity suggest that usability should be studied in two parts: objective and subjective usability. Our results from hypotheses H2a and H2b show that the relation among gamification, objective usability, and subjective usability of in-person counseling is established. Moreover, hypothesis H3b shows that the relationship between adoption and subjective usability of in-person counseling is established. This result indicates that subjective usability, or satisfaction, affects the adoption of in-person counseling.

Our findings on the moderated mediation effect of gamification and vividness of presentation on the objective and subjective usability of in-person counseling align with the findings obtained from a prior study, but are different from other prior studies [62,63,66,84]. For instance, Guadagno showed that vividness influences message success and could positively or negatively affect persuasion. However, Frey showed that vividness harms persuasion. Smith showed that vividness increases persuasion if the vivid information is consistent with the central theme of the message. Finally, Taylor showed that vividness increases persuasion if people are in a condition of greater attention. Here, our results concerning moderator mediation suggest that participants experiencing vividness of the presentation of gamification gave the highest subjective usability rating; at the same time, the participants gave the highest objective usability rating in a condition of a non-vividness of presentation under a gamification condition. Furthermore, the indirect effect of gamification on the adoption of in-person counseling through perceived subjective usability was highly significant in a vividness condition.

### 5.3. Practical Implications

Although people are resistant to in-person counseling, our study experimentally found that gamification has a positive effect on in-person counseling adoption [13]. As Sailer et al. argue, gamification has the primary purpose of promoting human motivation and performance in a particular activity [32]. The importance of gamification has been thoroughly discussed in an increasing number of health professions education research [81,82,83].

Organizations should carefully evaluate and select the objective and subjective usability priorities of young adult patients. Usability testing and its impact should be a major focus for many organizations, because usability testing can actually save time, money, and stress [85]. Unlike the arguments of previous works, considering the results of the proposed hypothesis, a sense of its subjective usability or satisfaction would persuade people to adopt in-person counseling [76,77]. The results of this study suggest that the difference between gamification’s effect on perceived objective and subjective usability of in-person counseling is small. However, when comparing the impact of both forms of usability on adoption rates, subjective usability was significant, while objective usability was almost negligible.

Organizations should also carefully consider when to use information with high vividness and when to use information with low vividness. Although past research has suggested that the impact of vividness is illusory or counterproductive, vividness has been shown to have a positive impact on recent innovation research [62,63,66,84]. This study suggests that high presentation vividness positively affects subjective usability and that the subjective usability (yes gamification and no gamification conditions) of in-person counseling was viewed very differently from when vivid presentation material was used. However, high presentation vividness negatively affects the perception of objective usability, and the perception of the objective usability (under yes gamification and no gamification conditions) of in-person consultation also shows a big difference. Therefore, we should also carefully evaluate and choose the vividness of a presentation. Because the essence of gamification is to facilitate the participants’ learning processes, we should not focus on the participant’s satisfaction at the expense of the ability to disseminate knowledge.

### 5.4. Limitations and Future Directions

Our research suffers from the following four limitations. When future researchers address them, they will extend and deepen the insights we obtained from our experiment.

First, our study concerns pre-process expectations and gamification’s effects on people’s adoption of in-person counseling, which is equivalent to people’s perceptions of and intentions about in-person counseling [86,87]. Therefore, the informational value of this work is limited to the pre-counseling phase and does not include the actual actions people take when accessing counseling, such as making an appointment, finding the right therapist, etc. as well as the people’s subsequent actions and experiences while experiencing counseling. However, this pre-counseling phase can be considered a critical one, as it influences people’s understanding of and resistance to counseling, and thus whether they will act on it in subsequent times.

Second, the principal limitation of the method we used is that there could have been many ways to introduce in-person consultation. However, due to limited professional background and resources, we only used the in-person counseling introduction video as a tool to conduct our study. Second, we selected two materials for comparing the vividness of the presentation. In the process of experimental comparison, we found that analysis of multiple materials may yield more understanding of the moderated mediation effect of the vividness of presentation, i.e., vividness can be divided into figural and ground vividness [63].

Third, in our experiments, we used the ISO 9241-11 standard for usability measurements [49] even though there are many commercially available measurement tools for usability, such as VISAWI, meCUE, UEQ, and other measurement tools. Therefore, we recommend that a follow-up analysis be performed in which measurements can be made using a different measurement tool.

Fourth, in our study, we have treated sex, age, and nation as independent covariates. However, more research is needed to understand the possible confounding effect of these three or more types of covariates. Further work should be done to understand groups other than young adults, such as adolescents or baby boomers [31]. In addition, further research is needed to examine other factors that influence contextual and demographic factors, such as educational background and family conditions. Therefore, we recommend a follow-up analysis that takes all of the above into account.

### 5.5. Conclusions

In contrast to most previous descriptive studies, our study is the first to uncover the association between gamification and adoption of in-person counseling using an experimental approach. Using a 2 × 2 between-subjects design, we investigated the relationship between gamification and adoption in young adults. We also identified underlying processes associated with adoption among young adults through the objective and subjective usability of in-person counseling and the vividness of the presentation. Our findings provide quantitative evidence of the effects of gamification on the adoption of in-person counseling.

## Figures and Tables

**Figure 1 behavsci-12-00040-f001:**
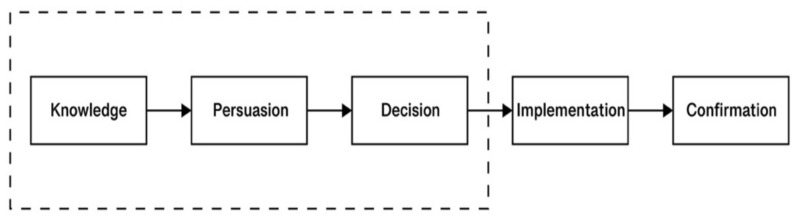
A Model of the Adoption Stage by Rogers.

**Figure 2 behavsci-12-00040-f002:**
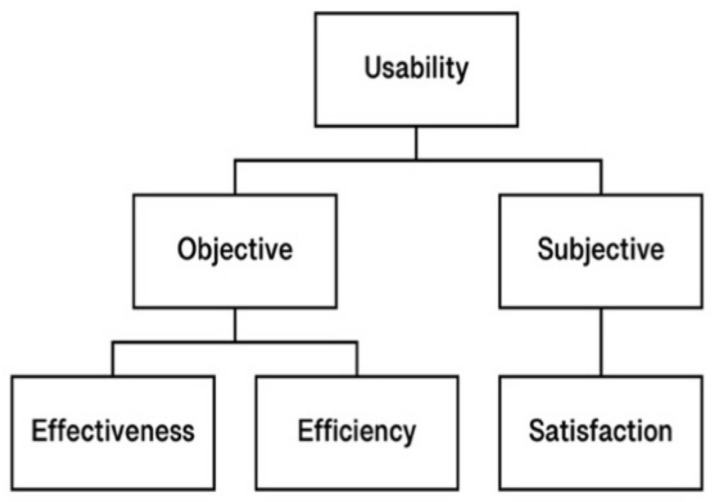
A Framework of Usability According to Yen’s Health IT Usability Evaluation Model (Health-ITUEM).

**Figure 3 behavsci-12-00040-f003:**
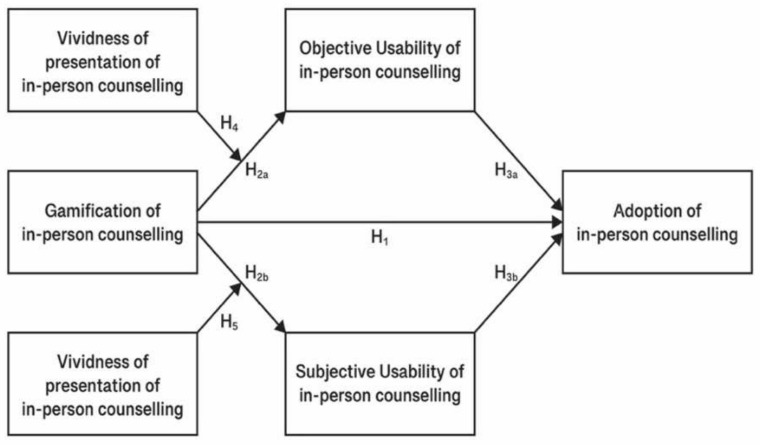
Adoption of in-person counseling as the function of the gamification of in-person counseling, the objective usability of in-person counseling, the subjective usability of in-person counseling, and the vividness of presentation of in-person counseling.

**Table 1 behavsci-12-00040-t001:** The difference between gamification: no and gamification: yes according to environment element.

	Gamification: No	Gamification: Yes
Environment	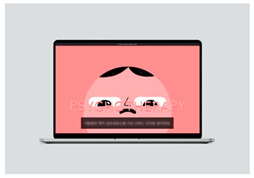	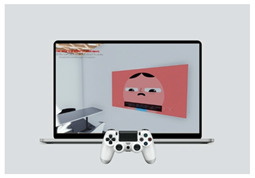

**Table 2 behavsci-12-00040-t002:** Stimuli differing 2 × 2 between-subjects design.

	Gamification: No	Gamification: Yes
Vividness: low	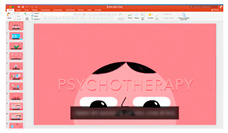	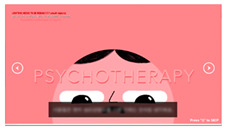
Vividness: high	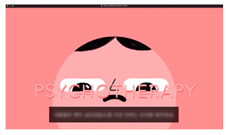	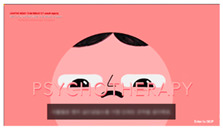

**Table 3 behavsci-12-00040-t003:** Participants’ characteristics.

Measure	Groups	N (%)	M (SD)
Sex	Male	42 (35%)	-
	Female	78 (65%)	
Age	20–29	67 (55.83%)	29.333 (4.198)
	30–40	53 (44.17%)	
Nationality	Korea	62 (51.7%)	-
	China	58 (48.3%)	

**Table 4 behavsci-12-00040-t004:** Correlation analysis among effectiveness, efficiency, and satisfaction.

Variable	1	2	3
1. Effectiveness	1.000		
2. Efficiency	0.666 ***	1.000	
3. Satisfaction	0.145	0.117	1.000
Mean	79.167	34.167	64.792
Std. Deviation	17.178	45.827	12.861

*** *p* < 0.001, ** *p* < 0.01, * *p* < 0.05.

**Table 5 behavsci-12-00040-t005:** Total Variance Explained.

Component	Eigenvalue	Difference	Proportion	Cumulative
1	1.714	0.762	0.571	0.571
2	0.953	0.619	0.318	0.889
3	0.333	-	0.111	1.000

**Table 6 behavsci-12-00040-t006:** The rotated component matrix between effectiveness, efficiency, and satisfaction.

Variable	Comp1	Comp2	Comp3	Unexplained
Effectiveness	0.913	0.045	-	0
Efficiency	0.908	0.087	-	0
Satisfaction	0.072	0.997	-	0

Notes. Extraction Method: Principal Component Analysis. Rotation Method: Varimax with Kaiser Normalization.

**Table 7 behavsci-12-00040-t007:** Statistical power of independent sample *t*-tests sample size.

Effect Size D	α Err Prob	Power (1-β Err Prob)	Noncentrality Parameter δ	Critical t	Recommended Total Sample Size	Actual Total Sample Size	Actual Power
0.830	0.100	0.800	2.560	1.688	38	120	0.807

**Table 8 behavsci-12-00040-t008:** Direct effects of gamification on adoption via objective and subjective usability of in-person counseling.

Output	Mediator	Model	R^2^	B	SE	t	*p*	LLCI	ULCI
Adoption	Objective usability	Gamification → Objective Usability (path a)	0.120 **	0.546	0.181	3.014	0.003	0.246	0.846
		Objective usability → Adoption (path b)	0.199 ***	−0.078	0.109	−0.712	0.478	−0.259	0.103
		Gamification → Adoption (path c)	0.084 *	.520	.225	2.314	0.022	0.147	0.892
		Gamification → Adoption (path c’)	0.199 ***	0.378	0.225	1.676	0.096	0.004	0.751
	Subjective usability	Gamification → Subjective usability (path a)	0.097 *	0.434	0.184	2.365	0.020	0.130	0.738
		Subjective usability → Adoption (path b)	0.199 ***	0.426	0.108	3.954	0.000	0.247	0.604
		Gamification → Adoption (path c)	0.084 *	0.520	0.225	2.314	0.022	0.147	0.892
		Gamification → Adoption (path c’)	0.199 ***	0.378	0.225	1.676	0.096	0.004	0.751

Note. The confidence level for all confidence intervals in the output is 90%; *** *p* < 0.001, ** *p* < 0.01, * *p* < 0.05.

**Table 9 behavsci-12-00040-t009:** Indirect effects of gamification on adoption via objective and subjective usability of in-person counseling.

Gamification (X)	Adoption (Y)	Bias-Corrected Bootstrap 90% CI			
Mediator (M)		Indirect effect	BootSE	BootLL	BootUL
Objective usability		−0.042	0.071	−0.168	0.066
Subjective usability		0.185	0.102	0.042	0.372

Note. The confidence level for all confidence intervals in the output is 90%. The bootstrap sample used for the percent bootstrap confidence interval is 5000.

**Table 10 behavsci-12-00040-t010:** Statistical power of mediation analysis sample size.

Effect Size f^2^	α err Prob	Power (1-β err Prob)	Noncentrality Parameter λ	Critical F	Recommended Total Sample Size	Actual Total Sample Size	Actual Power
0.157	0.100	0.800	10.389	2.039	66	120	0.803

**Table 11 behavsci-12-00040-t011:** Interaction effects of gamification and vividness on the objective and subjective usability of in-person counseling.

Output	Model	R^2^	B	SE	t	*p*	LLCI	ULCI
Objective usability	Gamification × Vividness → Objective usability	0.656 ***	−1.329	0.224	−5.926	0.000	−1.702	−0.957
Subjective usability	Gamification × Vividness → Subjective usability	0.167 **	1.079	0.349	3.089	0.003	0.499	1.658

Note. The confidence level for all confidence intervals in the output is 90%; *** *p* < 0.001, ** *p* < 0.01, * *p* < 0.05.

**Table 12 behavsci-12-00040-t012:** Conditional effects of vividness on the objective and subjective usability of in-person counseling.

Gamification (X)		Bias-Corrected 90% CI					
Mediator (M)	Vividness (W)	Conditional Effect	SE	t	*p*	LL	UL
Objective usability	0	1.130	0.156	7.257	0.000	0.872	1.388
	1	−0.200	0.165	−1.213	0.228	−0.473	0.073
Subjective usability	0	−0.075	0.242	−0.311	0.757	−0.477	0.327
	1	1.003	0.256	3.914	0.000	0.578	1.428

Note. The confidence level for all confidence intervals in the output is 90%.

**Table 13 behavsci-12-00040-t013:** Indirect effects of gamification on adoption via objective and subjective usability and the vividness of presentation of in-person counseling.

Gamification (X)	Adoption (Y)	Bias-Corrected Bootstrap 90% CI			
Mediator (M)	Vividness (W)	Indirect Effect	BootSE	BootLL	BootUL
Objective usability	0	−0.088	0.142	−0.327	0.137
	1	0.016	0.034	−0.023	0.082
Subjective usability	0	−0.032	0.108	−0.199	0.155
	1	0.427	0.166	0.178	0.732

Note. The confidence level for all confidence intervals in the output is 90%. The bootstrap sample used for the percent bootstrap confidence interval is 5000.

**Table 14 behavsci-12-00040-t014:** Moderated mediation using a two-way analysis of variance (ANOVA) and the moderated mediation relationships between gamification and the vividness of in-person counseling.

Output	Measure	Sum of Squares	df	Mean Square	F	*p*	Partial Eta Squared
Objective usability	Corrected Model	72.997 ^a^	3	24.332	61.355	0.000	0.613
	Intercept	0.000	1	0.000	0.000	1.000	0.000
	Gamification	7.286	1	7.286	18.371	0.000	0.137
	Vividness	52.826	1	52.826	133.203	0.000	0.535
	Gamification × Vividness	12.885	1	12.885	32.490	0.000	0.219
	Error	46.004	116	0.397			
	Total	119.000	120				
	Corrected Total	119.000	119				
Subjective usability	Corrected Model	11.877 ^b^	3	3.959	4.287	0.007	0.100
	Intercept	0.000	1	0.000	0.000	1.000	0.000
	Gamification	4.594	1	4.594	4.975	0.028	0.041
	Vividness	0.016	1	0.016	0.017	0.896	0.000
	Gamification × Vividness	7.267	1	7.267	7.869	0.006	0.064
	Error	107.123	116	0.924			
	Total	119.000	120				
	Corrected Total	119.000	119				

Note. ^a^ represents R Squared = 0.613 (Adjusted R Squared = 0.603), ^b^ represents R Squared = 0.100 (Adjusted R Squared = 0.077).

**Table 15 behavsci-12-00040-t015:** Moderated mediation using a two-way analysis of variance (ANOVA) and the interaction relationships between gamification and vividness of in-person counseling.

Mediator (M)	Gamification (X)	Vividness (W)	Mean	Std. Error	LBCI	UBCI
Objective usability	0	0	0.089	0.115	−0.101	0.280
		1	−0.582	0.115	−0.773	−0.392
	1	0	1.238	0.115	1.047	1.428
		1	−0.745	0.115	−0.935	−0.554
Subjective usability	0	0	0.039	0.175	−0.252	0.330
		1	−0.430	0.175	−0.721	−0.139
	1	0	−0.062	0.175	−0.353	0.229
		1	0.453	0.175	0.162	0.744

Note. The confidence level for all confidence intervals in the output is 90%.

**Table 16 behavsci-12-00040-t016:** Statistical power of two-way ANOVA sample size.

Output	Effect Size f^2^	α err Prob	Power (1-β err Prob)	Numerator Df	Noncentrality Parameter λ	Critical F	Recommended Total Sample Size	Actual Total Sample Size	Actual Power
Objective usability	0.530	0.100	0.800	1	6.730	2.975	24	120	0.805
Subjective usability	0.261	0.100	0.800	1	6.291	2.763	92	120	0.800

## Data Availability

The data collected and presented in this study are available upon request.

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
