# Peer review of "A Behavioral Strategy to Nudge Young Adults to Adopt In-Person Counseling: Gamification"

_behavsci, 2022, doi:10.3390/bs12020040_

Round 1

Reviewer 1 Report

The article "A Behavioral Strategy to Nudge Young Adults to Adopt In-Person Counseling: Gamification" is a deep empirical study. The relevance of the study is due to the fact that, according to the authors, "seeking professional help is not only complicated but also unnatural for people". Due to the COVID-19 counseling will likely become a hybrid offered online and offline, so gamification can really be seen as a new strategy to help young adults to adopt in-person counseling. "To test our hypothesis, we conducted an experiment by employing a between-subject design. More specifically, we developed a virtual world-based game of the Psychological Diagnostic Institute and collected participants’ responses toward the virtual game as well as in-person counseling", authors write. Authors present seven hypotheses regarding the gamification of in-person counseling. In total, 120 young adults from Korean and Chinese-speaking regions participated in this study. As a result of the study, the authors come to the conclusion that gamification positively affected the adoption of in-person counseling and objective and subjective usability. Furthermore, vividness moderated the relationship between gamification and both objective and subjective usability. This study is the first to uncover the association between gamification and adoption of in-person counseling using an experimental approach and provides quantitative evidence of the effects of gamification on the adoption of in-person counseling. However, some questions require clarification: who specifically is this study intended for and its results? What focus group are these results for? What practical impact can these results have on the studied situation? These questions do not affect the positive assessment of the article.

Author Response

Thank you for your helpful comments and suggestions. We have adopted your suggestions and have carefully revised and improved our manuscript. For more details, please refer to the response note.

Reviewer 2 Report

I think this is well-organized and tidy research. The topic is very interesting and the writing is easy to read and follow. I feel that it can be accepted for publication after minor reversion. The following is my minor comments:

(1) I suggest that the authors can directly show their contributions in introduction section.

(2) Ethical events should be mentioned in the method section.

(3) In section 4.1.5, why do you choose these analysis methods? You should justify them.

(4) In conclusions section, it is too simple. You have introduced three research questions in Introduction. So, you’d better to answering them in conclusion section. In addition, whether does your research have any limitations? Or, what should other do about future studies?

Author Response

Thank you for your helpful comments and suggestions. We have taken your suggestions into consideration and have carefully revised and improved our manuscript. For more details, please refer to the response note.

Reviewer 3 Report

The authors have investigated if gamification improves young people attitude toward in-person counseling. This is an interesting study; however, it has a small sample size. The data analyses and findings of this study are poorly reported and it is very difficult to recognize the value of this study.  

Abstract:

  • “Mental illness has always been an important issue for young adult. Moreover, the outbreak of COVID-19 has amplified this problem. However, misunderstandings about in-person counseling and the gaze of others have led to resistance to in-person counseling.” These sentences need to be revised.
  • “One hundred twenty young adults participated in our experiment” please report the gender and mean age of the participants.

Introduction:

  • “Mental illness is a widely prevalent and critical issue for humanity.” This sentence should be removed as it does not add anything to the text.

Methods:

  • In the “data analysis” section, instead of referring to the hypotheses as for example “H1” briefly describe the hypothesis.
  • The authors should clearly describe their statistical models in the data analysis section.
  • Considering the small sample size of this study, the authors should calculate the statistical power and include its results in the results section.

General:

  • This manuscript would benefit from English editing by a subject-matter expert.

Author Response

(The authors gave the same response as above.)

Round 2

Reviewer 3 Report

I have no further comments/suggestions.